# Comparative Pan-Genome Analysis of Oral *Veillonella* Species

**DOI:** 10.3390/microorganisms9081775

**Published:** 2021-08-20

**Authors:** Izumi Mashima, Yu-Chieh Liao, Chieh-Hua Lin, Futoshi Nakazawa, Elaine M. Haase, Yusuke Kiyoura, Frank A. Scannapieco

**Affiliations:** 1Department of Oral Medical Science, School of Dentistry, Ohu University, Fukushima 963-8611, Japan; y-kiyoura@den.ohu-u.ac.jp; 2Institute of Population Health Sciences, National Health Research Institutes, Zhunan 35053, Miaoli Country, Taiwan; jade@nhri.edu.tw (Y.-C.L.); mammer@nhri.edu.tw (C.-H.L.); 3Department of Oral Biology, Faculty of Dentistry, Universitas Indonesia, Jakarta 10430, Indonesia; f.nakazawa.gm@gmail.com; 4Department of Oral Biology, School of Dental Medicine, University at Buffalo, The State University of New York, Buffalo, NY 14214, USA; haase@buffalo.edu (E.M.H.); fas1@buffalo.edu (F.A.S.)

**Keywords:** oral *Veillonella*, pan-genome analysis, BPGA, KEGG, metabolic pathways, lactate metabolism, fructose metabolism

## Abstract

The genus *Veillonella* is a common and abundant member of the oral microbiome. It includes eight species, *V. atypica*, *V. denticariosi*, *V. dispar*, *V. infantium*, *V. nakazawae*, *V. parvula*, *V. rogosae* and *V. tobetusensis*. They possess important metabolic pathways that utilize lactate as an energy source. However, the overall metabolome of these species has not been studied. To further understand the metabolic framework of *Veillonella* in the human oral microbiome, we conducted a comparative pan-genome analysis of the eight species of oral *Veillonella*. Analysis of the oral *Veillonella* pan-genome revealed features based on KEGG pathway information to adapt to the oral environment. We found that the fructose metabolic pathway was conserved in all oral *Veillonella* species, and oral *Veillonella* have conserved pathways that utilize carbohydrates other than lactate as an energy source. This discovery may help to better understand the metabolic network among oral microbiomes and will provide guidance for the design of future *in silico* and *in vitro* studies.

## 1. Introduction

Members of the genus *Veillonella*, belonging to the family *Veillonellaceae*, are strictly anaerobic Gram-negative cocci mostly isolated from the oral cavity and gut of mammals [1]. The genus *Veillonella* includes 15 recognized species [2]. With the exception of *Veillonella criceti*, *V. ratti* and *V. seminalis*, they appear unable to ferment carbohydrates or amino acids [3,4,5]. Alternatively, *Veillonella* have been shown to ferment short-chain organic acids, especially lactate, as a source of energy, and subsequently transform it to acetate and propionate [1,6,7,8]. Regarding the unique physiology of these species, it was recently reported that the relative abundance of *Veillonella* in the gut is significantly associated with increased performance in marathon running [9]. This mechanism was proved when *V. atypica* gavage improved treadmill run time in mice. The proposed mechanism for this remarkable finding is that serum lactate that entered the gut lumen was transformed by *V. atypica* to acetate and propionate, which allowed the mice to improve treadmill run time [9].

The oral *Veillonella* include the species *V. atypica*, *V. denticariosi*, *V. dispar*, *V. infantium*, *V. nakazawae*, *V. parvula*, *V. rogosae* and *V. tobetusensis* [10,11,12,13,14,15,16]. Knowledge of the ecology of oral *Veillonella* has improved over recent years. Culture studies have revealed that the distribution and frequency of oral *Veillonella* species in the oral cavity differs by surface. While the species *V. rogosae*, *V. atypica* and *V. dispar* are numerous on the tongue [17,18], *V. parvula* prefers the subgingival plaque [19,20]. In addition, *V. parvula* has been associated with periodontitis and dental caries, including severe early childhood caries [12,19,21]. Moreover, several salivary and plaque microbiome studies have revealed that higher proportions of *Veillonella* species are associated with periodontitis and dental caries [22,23,24]. Our previous microbiome study found that the proportion of *Veillonella* species increased with poor oral hygiene status in healthy subjects [25]. Furthermore, it has been suggested that *Veillonella* may be anti-cariogenic, since these species consume lactate, which is a major driver of dental caries [26]. *Veillonella* are frequently detected in high numbers in patients with active carious lesions [12,19,21,23,24]. Moreover, it has also been reported that *Veillonella* have a central role in early-stage biofilm formation together with *Streptococcus* species [27,28,29].

Recently, reports using in silico analysis of the genome of *V. parvula* and *V. atypica* [30,31], including *V. parvula* outer membrane proteins [32], have improved understanding of the metabolic and physiologic activities of this important genus. Pan-genome analysis represents a new approach to define the total species metabolic and physiologic capabilities of a genus or a species, and can provide a framework for estimating and/or modeling genomic diversity, and identifying core genomes (shared by all strains), accessory genomes (dispensable genes existing in two or more strains), and unique genes (specific to a single strain) [33]. The core genome is the essence of a phylogenetic unit, and it is thought to be representative of a taxon [34]. The accessory genome, on the other hand, includes key genes needed to survive in specific environments; it is commonly linked to virulence, capsular serotype, adaptation, and antibiotic resistance and might reflect the organism’s unique characteristics [35]. Such unique genes, as evidenced for example in *Streptococcus agalactiae* [36], are clustered in genomic islands. They are often flanked by insertion elements and display an atypical nucleotide composition, suggesting that their acquisition occurred through horizontal transfer. These findings increase the understanding of genetic differences and related functions of a study group of organisms.

The aim of the present study was to perform comparative pan-genome analysis to identify differences in functional gene distribution among draft or complete genomes of all eight species of oral *Veillonella*, and to understand their potential functions that allow them to adapt to the complex oral environment. Specifically, we focused on glycolysis and related pathways conserved in all oral *Veillonella* to better understand carbohydrate metabolism of the genus *Veillonella*.

## 2. Materials and Methods

### 2.1. Bacterial Strains and Growth Conditions

The strains of oral *Veillonella* included in this study are listed in Table 1. *V. atypica*, *V. denticariosi*, *V. infantium* and *V. rogosae* were cultured on Bacto^TM^ Brain Heart Infusion (Difco Laboratories BD) agar supplemented with 5% (volume/volume) defibrinated sheep blood and incubated under anaerobic conditions (N_2_:H_2_:CO_2_ = 80:20:20) at 37 °C for 5 days prior to DNA isolation.

### 2.2. Draft or Complete Genome Sequences, Assemblies and Annotation

Genomic DNA was extracted from each strain using the phenol-chloroform extraction and ethanol precipitation procedures [37] and further purified using the QIAamp DNA minikit (Qiagen) for high-throughput sequencing according to the manufacturer’s instruction. DNA library preparation, DNA sequencing, de novo assembly and annotation of these four strains were already reported [38]. Briefly, DNA libraries were prepared using the Nextera DNA library preparation kit (Illumina). DNA sequencing was performed using the Illumina NextSeq 500 analyzer for paired-end sequences. The paired-end sequencing reads were checked for quality, de novo assembled, and annotated using MyPro, a software pipeline for prokaryotic genomes [39].

### 2.3. Comparative Pan-Genome Analysis of Oral Veillonella

The pan-/core-genome analysis of eight *Veillonella* genomes (Table 1) was carried out by Bacterial Pan Genome Analysis tool (BPGA) pipeline v1.3 [33], using each nucleotide sequence (gbk file) with the default setting. In BPGA pipeline, orthologous protein clusters were identified with USEARCH [40] using a threshold of 0.5. The core and accessory protein families identified by the BPGA pipeline were then used to perform KEGG pathways [41]. Enrichment analysis was conducted using the R package clusterProfiler, with a Benjamini–Hochberg correction. The pathway with the corrected *p*-value <0.05 and *q*-value <0.2 were considered significantly enriched [42]. A syntonic analysis was performed by Mauve [43] with default setting to investigate the locally collinear blocks (LSBs) conserved among eight *Veillonella* species.

## 3. Results and Discussion

### 3.1. Pan-Genome Construction

We used both complete and draft genomes (Table 1), an approach that has been used in previous studies [44,45,46]. The genomic features of all eight oral *Veillonella* species used in this study are shown in Table 1. Additionally, detailed descriptions of the complete or draft genomes of six of the eight *Veillonella* oral species were previously reported [38,47,48].

### 3.2. Pan-Genomic Analysis

The core genomes, accessory genomes and unique genes in all eight oral *Veillonella* species were generated by BPGA and are shown in Figure 1a. The largest number of accessory genes was 450, which were found in *V. rogosae*, while the smallest number of accessory genes was 287, in *V. denticariosi*. These results suggest that characteristics of *V. rogosae* showed the predominant characteristics of oral *Veillonella*. *V. rogosae* has been isolated and detected frequently in the oral cavity and is considered to be a predominant species among oral *Veillonella* [17,18]. On the other hand, *V. tobetusensis* was found to have the largest number of unique genes among oral *Veillonella* (Figure 1a). At the same time, *V. atypica* had the largest number of the atypical GC content (45.1%) in unique genes among the eight species, and *V. tobetuensis* had the smallest one (14.5%) (Appendix A). According to these results, *V. atypica* may accept exogeneous genes easily by lateral gene transfer, and *V. tobetuensis* may show the unique characteristics compared to other oral *Veillonella* species.

Moreover, in the pan and core genome plot (accumulation curve) of oral *Veillonella* (Figure 1b), the size of the pan-genome increases on addition of each genome, whereas the core genome reduces with the addition of every new genome, suggesting an “open” pan-genome. The total gene families in Figure 1b is the sum of gene families not found in any of the previous genomes (Figure 1c). The gene family frequency spectrum (Figure 1d) presents the number of unique and core, accessory gene families, 837 unique gene families that present in only one genome and 1325 core genome families that present in the eight genomes, and the others are accessory gene families that present in 2–7 genomes of the eight genomes. The distribution of the core and accessory genome, and unique genes revealed that the genomes of oral *Veillonella* species were remarkably diverse.

### 3.3. COG Distribution of Core, Accessory Genome and Unique Genes

A search for core, accessory and unique gene families were conducted to compare the distribution of functional categories by using Clusters of Orthologous Groups of proteins (COGs) database [49] through BPGA [33]. Figure 1e shows the differential distribution of COG functional categories in core, accessory, and unique gene families. The most common functions (44.0%) in the core genomes of oral *Veillonella* species are associated with metabolism (Figure 1e). Class E (Amino acid transport and metabolism) was the most enriched (10.5%) metabolic function. Meanwhile, class J (Translation, ribosomal structure and biogenesis) belonging to cellular processing and signaling functions showed almost the same degree of enrichment (10.6%) with class E in the core genomes. According to the result of the COG distribution, the majority of genes belonging to the core group were related to housekeeping functions. Additionally focused on the accessory genome in class E and class J, class J genes were more conserved in oral *Veillonella* species (2.00% of that accessory genome), while class E genes comprised 10.6% of that group. It was suggested that class E genes might suggest the different abilities depending on the species of oral *Veillonella*. Likewise, when comparing the COG groups for metabolism, the percentage of class P (Inorganic ion transport and metabolism) genes was variable and were found in higher fractions (10.6%) of the accessory genome versus core genome (6.2%). On the contrary, class C (Energy production and conversion) and class H (Coenzyme transport and metabolism) genes were relatively conserved in oral *Veillonella*, which were 7.0% and 7.1% in the core genome versus 3.5% and 3.9% in the accessory genome.

About 20% of the core genome content was grouped under class R (General function prediction only) and class S (Function unknown) having poorly characterized function. Likewise, among the genes from the accessory genome and unique genes, approximately 26.7–27.8% of the total gene content was grouped under the COG same classes, with no specific function assigned to these genes. The oral *Veillonella* have potential pathways or abilities not yet estimated by the present COG categories.

### 3.4. Phylogenetic and Evolutionary Analysis of Oral Veillonella

BPGA generated three phylogenetic trees, concatenated core gene alignments, and using pan-genome and accessory genomes, respectively (Figure 2) [40]. According to results of phylogenetic analysis of oral *Veillonella* species, the pan-phylogenic tree (Figure 2a) showed complex branches compared to the core-phylogenic tree (Figure 2b). In addition, an accessory-phylogenetic tree showed similar clusters as observed with the pan-phylogenetic tree (Figure 2a,c). This result suggested that the construction of the pan-phylogenetic tree was influenced by accessory genomes and contained key genes of phylogenetic or evolutionary significance among oral *Veillonella* species. Furthermore, according to the results of our recent study, *V. nakazawae* was closely related to *V. infantium* and *V. dispar*, based on analysis of several house-keeping gene sequences including 16S rRNA [16]. Here, the core- and accessory-phylogenetic tree supported the relationship of these three species better than the pan-phylogenetic tree, since the COG distribution of the core genome was related to house-keeping functions in oral *Veillonella*, and accessory genomes had equivalent key functions in these species. It was suggested that the unique genes might enrich their evolutionary lineage in the pan-phylogenetic tree.

To study the evolutionary context in more detail, a syntonic analysis among the eight oral *Veillonella* species was also performed. The analysis showed that the eight genomes of oral *Veillonella* seemed similar in content, but not similar in gene synteny (Appendix A). The syntonic map depicted linearized alignments identifying about 120 conserved gene regions, however, it was hard to understand these alignments among the eight species. According to these results, a syntonic analysis will be required among strains of each species to identify the specific locally collinear blocks clearly and the comparison of one for one species genome, consequently the genome comparison among eight species should be analyzed in the future studies.

### 3.5. KEGG Pathway Mapping of Genes

The pan-genome functional analysis module of BPGA was also used for KEGG (Kyoto Encyclopedia of Genes and Genomes) pathway mapping of representative protein sequences of core, accessory genomes and unique genes of oral *Veillonella*. Appendix A listed all countable KEGG pathways with the KEGG major and sub-categories in oral *Veillonella* where at least one gene was detected. According to the Appendix A information, it was suggested that various pathways might be conserved in oral *Veillonella* to adapt to the oral environment. In addition, these pathways might vary by accessory and unique genes. KEGG assignments from BPGA revealed overall higher representation of metabolism related pathways (Figure 3a). This result also strongly supported the result of the COG distribution regarding metabolic function. The Histidine metabolism pathway (part of Histidine biosynthesis pathway) was composed of a core and nine accessory genomes (Appendix A). This was the only pathway predominantly constructed with accessory genomes, suggesting that Histidine biosynthesis might be the representative of all oral *Veillonella*.

An interesting finding of potential pathogenic traits for oral *Veillonella* species, the bacterial secretion system belonging to the KEGG major category of Environmental Information Processing was conserved with 12 core genes, two accessory genomes and a unique gene (Appendix A). This system was identified as the incomplete type II secretion system (T2SS) among the eight oral *Veillonella*. Knapp et al. reported that they identified the likely source of DNA uptake machinery within a locus homologous to T2SS in *V. parvula* [30]. Our results supported their results and added additional evidence at the genetic level to determine T2SS in genus *Veillonella* as a potential pathogenic trait.

In the sub-category of metabolism, the most abundant function in the core genomes conserved carbohydrate metabolism (Figure 3b). Furthermore, according to the results of enrichment analysis in core genomes, four KEGG pathways, carbon metabolism, ribosome, biosynthesis of amino acids and biosynthesis of cofactors, were specifically enriched (more than 5% with enrichment significance) (Figure 4a). Regarding accessory genomes, pathways of carbon metabolism and ribosomes were not found (Figure 4b), suggesting that these two KEGG pathways were more conserved in oral *Veillonella* species.

### 3.6. Glycolysis and Its Related KEGG Pathways in Carbon Metabolism of Oral Veillonella

In this study, pathways related to carbohydrate metabolism in KEGG pathways were conserved in oral *Veillonella*. Regarding carbohydrate metabolism, five KEGG pathways related to glycolysis were investigated. Figure 5 shows five integrated metabolic pathways found in all oral *Veillonella* species. As already reported, these pathways consume lactate as a source of energy, and subsequently transform it to acetate and propionate [1,6,7,8]. A metabolic pathway was conserved in all oral *Veillonella* species (Figure 5). It appears that an incomplete TCA cycle was used from pyruvate to malate, fumarate, succinate and succinyl-CoA for production of propionate (Figure 5). Additionally, it is known that a specific malic-lactic transhydrogenase catalyzed the reaction, the conversion of lactate and oxaloacetate to malate and pyruvate [7]. Malic-lactic transhydrogenase, which catalyzes a directed transfer of reducing equivalents from lactate to oxaloacetate to form malate, affects a sparing of electrons derived from the ferredoxin-mediated phosphoroclastic decarboxylation of pyruvate [51]. In a study of *V. parvula* M4, Ng and Hamilton [52] observed the inability of cell-free extracts of this organism to metabolize lactate in the absence of oxaloacetate and detected by this experimentation the direct coupling of lactate dehydrogenation to the presence of oxaloacetate, thus establishing the presence of malic-lactic transhydrogenase in *V. parvula* M4. However, in this pan-genomic study, this specific enzyme was not mapped in any oral *Veillonella* species. As the gene or protein of malic-lactic transhydrogenase has not been reported, the KEGG database does not include this information. We speculate that the protein information of malic-lactic transhydrogenase was distributed in classes R or S in COG categories in this study (Figure 1e). However, according to the result of this analysis, lactate consumption without malic-lactic transhydrogenase is possible (Figure 5).

Moreover, we found that the metabolic pathway for fructose consumption was conserved in all oral *Veillonella* (Figure 5). In this pathway, fructose is consumed through the EMP pathway and is subsequently transformed to acetate and propionate. This result strongly suggests that oral *Veillonella* could utilize fructose as a source of energy beside lactate. Until now, it was suggested that they could utilize lactate and convert it to fructose to make several essential materials, like UDP-glucose. However, this is the first report for oral *Veillonella* of a conserved pathway that could utilize fructose as a nutrient source. This discovery also verified previous reports of fructose consumption by *V. seminalis* isolated from human clinical samples of semen [4].

Interestingly, the *pgi* gene classified to accessory genome was not found only in *V. dispar* (Figure 5). This gene of *V. dispar* might affect its phylogenetic lineage (Figure 2). In addition, the converting enzyme between Methyl-malonyl-CoA and Propanoyl-CoA was not mapped in this study (Figure 5). Perhaps an alternative pathway is present, as this process is essential to propionate production as a metabolic end product for all oral *Veillonella*.

## 4. Conclusions

The genus *Veillonella* is an important constituent of the oral microbiome [22,23,24,25]. This work represents the first characterization of all oral *Veillonella* species using pan-genomic analysis. Specifically, the discovery of the conserved pathway of fructose metabolism in all members of oral *Veillonella* increases understanding of the metabolic network within the oral microbiome that influences oral biofilm formation, along with *Streptococcus* species, as initial colonizers of the teeth. Moreover, these results may impact understanding of the pathogenesis of dental caries and periodontitis. The detail functions of such pathways need verification by in vitro studies to understand fructose metabolism and identification of intermediate metabolites by metabolome analysis. Finally, the results of this study increase our understanding of the characteristics of oral *Veillonella* and will facilitate future studies of this genus.

## Figures and Tables

**Figure 1 microorganisms-09-01775-f001:**
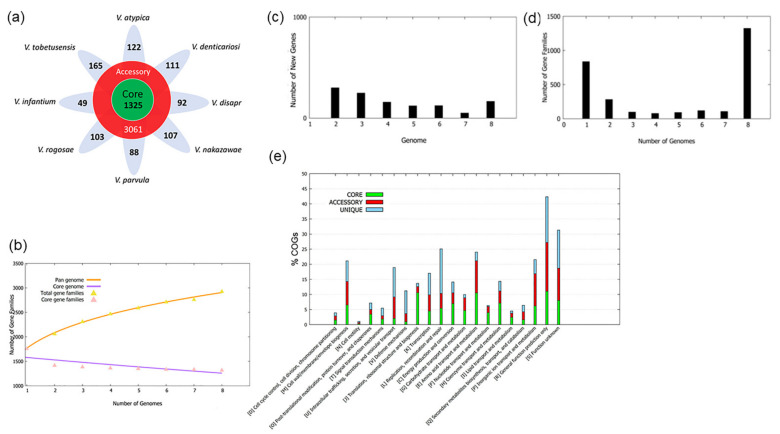
Genetic diversity in oral *Veillonella*. Each number of genomes represents organism names: 1. *V. atypica*, 2. *V. denticariosi*, 3. *V. dispar*, 4. *V. nakazwae*, 5. *V. parvula*, 6. *V. rogosae*, 7. *V. infantium*, 8. *V. tobetsuensis*. (**a**) Core, accessory and unique gene families of the eight type strains of oral *Veillonella*. The number of core genomes shared by all strains is in the center (1325) using green color. The total number of accessory genomes shared by two-seven strains is shown in red color circle (3061). The number of non-overlapping portions of each oval represents the size of unique genes shown in light blue. (**b**) A pan and core genome plot of oral *Veillonella*. The plot shows how the number of gene families increase and decline in the pan and core genome with each consecutive addition of a *Veillonella* genome. (**c**) New genes distribution after sequential addition of each genome to the analysis. (**d**) The gene family frequency spectrum. (**e**) COG distribution of core, accessory and unique genes. Classes D, M, N, O, T, U and V belonging to the category of Cellular processing and signaling. Classes J, K, and L belonging to the category of Information storage and processing. Classes C, G, E, F, H, I, Q and P belonging to the category of Metabolism. Classes R and S belonging to the category of Poorly characterized.

**Figure 2 microorganisms-09-01775-f002:**
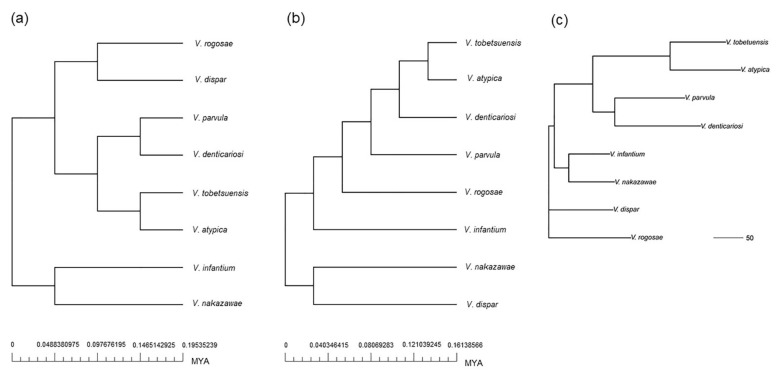
Phylogenetic analysis by BPGA using eight strains of oral *Veillonella*. A time scale is depicted in millions of years ago (MYA). (**a**) Pan-genome, (**b**) core genomes, (**c**) accessory genomes: this phylogenetic tree was constructed by using the binary matrix presented accessory gene presence/absence (1/0) in each genome. Subsequently, the neighbor-joining (NJ) method was used for the accessory genome binary matrix by ape package v5.5 [50]. The scale bar of NJ tree represents the genetic distance. The distance between two genomes has the number of loci for which they differ, and the associated variance is d (L-d)/L, where L means the number of loci.

**Figure 3 microorganisms-09-01775-f003:**
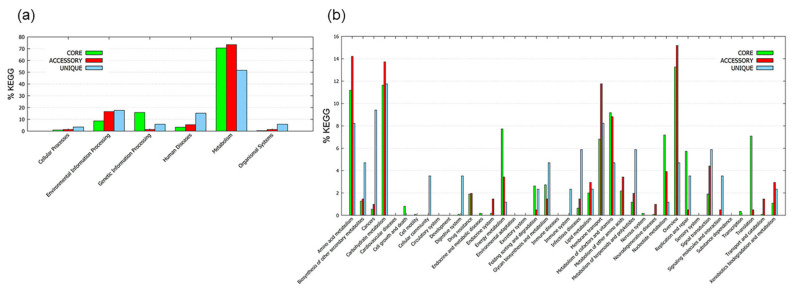
KEGG distribution of core, accessory and unique genes. (**a**) Distribution in major category. (**b**) Distribution in sub-category.

**Figure 4 microorganisms-09-01775-f004:**
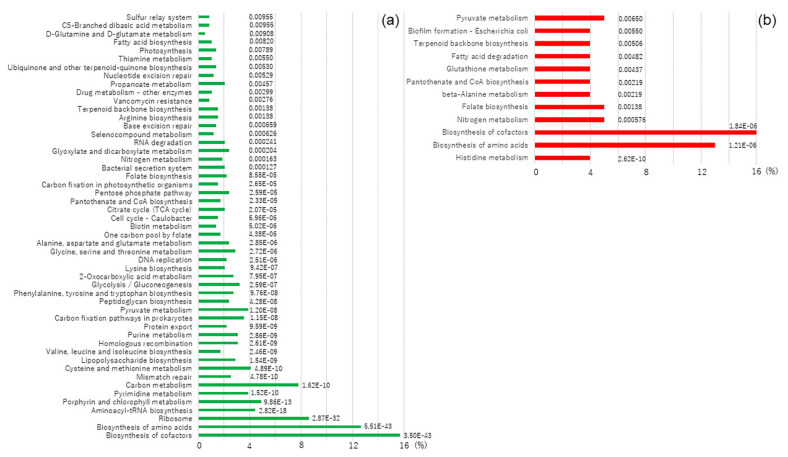
Enrichment results of core and accessory genomes. The enrichment significance was shown as the numerical values next to each bar graph. (**a**) Core genomes using the enrichment analysis. (**b**) Accessory genomes using the enrichment analysis.

**Figure 5 microorganisms-09-01775-f005:**
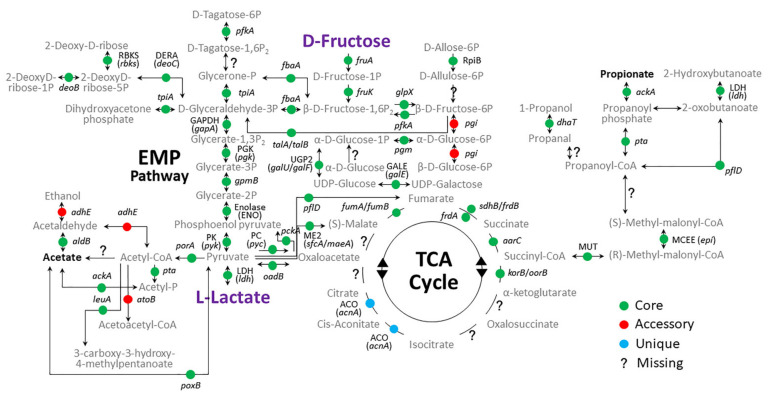
Incomplete pathway map integrated five KEGG pathways related to carbohydrate metabolism in oral *Veillonella*. Five KEGG pathways, 00010 Glycolysis/Gluconeogenesis, 00020 Citrate cycle (TCA cycle), 00051 Fructose and mannose metabolism, 00620 Pyruvate metabolism and 00640 Propanoate metabolism were partially integrated based on the mapping of the core, accessory and unique genes related to carbohydrate metabolism. This pathway is shown with intermediate metabolites and genes that are represented as core genes (green circles), accessory genes (red circles) and unique genes (light blue circles). Question marks in this pathway means the missing enzymes or genes that were not identified in this analysis.

**Table 1 microorganisms-09-01775-t001:** The genome information assembled in this study. The protein sequences of the four *Veillonella* strains, *V. dispar*, *V. nakazawae*, *V. parvula* and *V. tobetsuensis* were collected from National Center for Biotechnology Information. The others were assembled in this study.

Genome No.	Species Name	Strain	Type Strain	Assembly Level	Genome Size (bp)	N50	G+C (%)	Number of Genes	Number of CDSs	Number of Proteins	Data Source of Nucleotide Sequence	Accession Numbers
1	*Veillonella atypica*	ATCC 17744	YES	Draft	2,037,410	300,566	39.0	1928	1864	1832	https://ftp.ncbi.nlm.nih.gov/genomes/all/GCA/002/959/915/GCA_002959915.1_ASM295991v1 (accessed on 22nd July 2021)	PPDE01000000
2	*Veillonella denticariosi*	JCM 15641	YES	Draft	1,981,866	600,371	42.9	1852	1783	1746	https://ftp.ncbi.nlm.nih.gov/genomes/all/GCA/002/959/855/GCA_002959855.1_ASM295985v1 (accessed on 22nd July 2021)	PPDB00000000
3	*Veillonella dispar*	ATCC 17748	YES	Draft	2,116,567	498,249	38.9	1991	1926	1903	https://ftp.ncbi.nlm.nih.gov/genomes/all/GCF/000/160/015/GCF_000160015.1_ASM16001v1/ (accessed on 5th September 2017)	NZ_ACIK00000000
4	*Veillonella nakazawae*	JCM 33966	YES	Complete	2,097,818	2,097,818	38.6	1957	1893	1925	https://ftp.ncbi.nlm.nih.gov/genomes/all/GCA/013/393/365/GCA_013393365.1_ASM1339336v1/ (accessed on 8th July 2020)	AP022321
5	*Veillonella parvula*	DSM 2008	YES	Complete	2,132,142	2,132,142	38.6	1904	1840	1824	https://ftp.ncbi.nlm.nih.gov/genomes/all/GCF/000/024/945/GCF_000024945.1_ASM2494v1/ (accessed on 5th September 2017)	NC_013520.1
6	*Veillonella rogosae*	JCM 15642	YES	Draft	2,187,106	175,154	38.9	2068	2002	1951	https://ftp.ncbi.nlm.nih.gov/genomes/all/GCA/002/959/775/GCA_002959775.1_ASM295977v1 (accessed on 22nd July 2021)	PPCX00000000
7	*Veillonella infantium*	JCM 31738	YES	Draft	2,021,343	235,046	36.6	1899	1837	1809	https://ftp.ncbi.nlm.nih.gov/genomes/all/GCA/002/959/895/GCA_002959895.1_ASM295989v1 (accessed on 22nd July 2021)	PPDD00000000
8	*Veillonella tobetuensis*	ATCC BAA-2400	YES	Draft	2,161,277	225,588	38.5	2018	1948	1896	https://ftp.ncbi.nlm.nih.gov/genomes/all/GCF/001/078/375/GCF_001078375.1_ASM107837v1/ (accessed on 5th September 2017)	NZ_BBXI00000000

## Data Availability

The sequence data presented in this study are openly available with accession numbers as shown in Table 1. The nucleotide sequences used for BPGA are available in https://www.dropbox.com/sh/nh294a66dk9mkgi/AABSktK9WTDeNNsLDhnYYhlQa/gbk_rename.allGenBank?dl=0&subfolder_nav_tracking=1.

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
