# Peer review of "Comparative Pan-Genome Analysis of Oral Veillonella Species"

_microorganisms, 2021, doi:10.3390/microorganisms9081775_

Round 1
Reviewer 1 Report
Mashima et al. compared the genomes of eight Veillonella species for pan-genomic analysis, including V. atypica, V. denticariosi, V. dispar, V. infantium, V. nakazawae, V. parvula, V. rogosae, and V. tobetusensis. These genome sequences were obtained by DNA sequencing or downloaded from public databases. Veillonella is a common oral bacteria that is closely related to Streptococcus during the formation of oral biofilms and therefore is related to the oral microbiome. Veillonella's genome analysis and comparison are very important and will provide useful information to understand the dynamic changes of the oral microbiome. The study provides metabolic information by COG and KEGG analysis. The analysis showed that in addition to lactic acid metabolism for carbohydrate utilization, there are also conservative fructose metabolism pathways among all 8 Veillonella species. In general, the information presented in this manuscript is supported by reasonable statements and data. However, the following points should be addressed, and more analysis is needed to reveal biological functions.
Major comments:
- For uncompleted genomes, N50 data should be included to estimate the level of scaffold assembly.
- A syntonic analysis should be included for collinear blocks conserved among eight species to study the evolutionary context in more detail.
- Should there be any atypical GC content areas where HGT is present?
- More importantly, pan-genome comparisons can not only provide core or pan-genomes but also better understand the genotype-phenotype associations of specific biological groups. There is no biological information or discussion of the ecological effects of different Veillonella species through analysis of accessory genomes.
Author Response
Thank you for your helpful comments. We revised our manuscript according to your comments and suggestions. Please see attached.

Reviewer 2 Report
The manuscript “Comparative pan-genome analysis of oral Veillonella species for profiling genetic characteristics” deals with a comparative pan-genome analysis of the eight species of oral Veillonella. The article is well written and each analysis is well determined, but more discussion of the results should be considered. I suggest to address more the study on the pangenomic data, not only listing the fructose and lactate conserved metabolic pathways, but deeply investigating the core, singletons and accessory genes list in order to understand at genomic level the differences of this species. Furthermore, I strongly suggest to use another way to visualize table 2 (e.g. heat-map) or to include it as supplementary materials.
Author Response

(The authors gave the same response as above.)

Round 2
Reviewer 1 Report
All my previous questions have been answered properly. I have no further questions.
Reviewer 2 Report
The manuscript can be accepted in this form.